# A Study of the Reliability and Accuracy of the Real-Time Detection of Forage Maize Quality Using a Home-Built Near-Infrared Spectrometer

**DOI:** 10.3390/foods11213490

**Published:** 2022-11-03

**Authors:** Fei Gao, Yuejing Zhang, Xian Liu

**Affiliations:** 1China-Canada Joint Lab of Food Nutrition and Health (Beijing), School of Food and Health, Beijing Technology and Business University, 11 Fucheng Road, Beijing 100048, China; 2Biomass and Bioresource Utilization Laboratory, College of Engineering, China Agricultural University, Beijing 100083, China

**Keywords:** forage maize, influence factor optimization, online near infrared spectroscopy, model establishment and validation

## Abstract

The current study was conducted to explore the real-time detection capability of a home-built grating-type near-infrared (NIR) spectroscopy online system to determine forage maize quality. The factor parameters affecting the online NIR spectrum collection were analyzed, and the results indicated that the detection optical path of 12 cm, conveyor speeds of 10 cm s^−1^, and number of scans of 32 were the optimal parameters. Choosing the crude protein and moisture of forage maize as quality indicators, the reliability of the home-built NIR online spectrometer was confirmed compared with other general research NIR instruments. In addition, an NIR online multivariate analysis model developed using the partial least squares (PLS) method for the prediction of forage maize quality was established, and the reliability, applicability, and stability of the NIR model were further discussed. The results illustrated that the home-built grating-type NIR online system performed satisfying and comparable accuracy and repeatability of the real-time prediction of forage maize quality.

## 1. Introduction

Maize is an abundant and important crop in the world and is widely consumed as an important animal feed source. The utilization of maize in compound feed is 50–70%, and the total consumption of forage maize will continue to grow [1,2]. In addition, global demand for livestock products has continuously increased in recent decades, as the consumption of meat has increased in many countries [3,4]. It is noteworthy that the quality of forage maize directly affects livestock health and animal products such as meat, eggs, and milk. Therefore, the control of forage maize quality is of importance to ensure animal product safety and even human health [5].

Near-infrared (NIR) is the region in which absorptions corresponding to overtones and combinations of the fundamental vibrational transitions occur [6,7]. In recent decades, with the development of new NIR technologies, NIR online analytical technology has been used in agriculture, food, medicine, tobacco, and other industries as a rapid, efficient, and effective tool for the process of real-time analysis [6,8,9,10,11,12,13,14]. The advantages of NIR spectroscopy over traditional methods are speed, accuracy and precision, no sample destruction, simple sample preparation, the ability to provide simultaneous information on several properties and constituents, relatively reduced operational costs, and no use of toxic reagents [8,15]. Combined with the fiber optic probe, non-invasive NIR online measurements can be carried out from the production of raw materials to the quality control in production processes, as well as the final quality analysis of products [16].

However, most feed quality detection in practical production have mainly relied upon the laboratory NIR spectrometer, which cannot satisfactorily guide the real-time production process. Therefore, for industrial applications, realizing online detection in real-time is essential for the rapid development of the modern feed industry, which provides the ability to detect the raw material, process control, and end product at the same time. Online detection also improves the analysis efficiency and prompt feedback for process control [17].

Actually, the use of the NIR technique for online process monitoring is still limited. The stability of the spectral collection is influenced by the sample state or the transport environment in the actual production process [18]. In addition, the online spectra acquisition compared with the stationary measurement may result in a relatively low signal-to-noise ratio of spectra due to the high noise generated by the mechanical vibrations of the collection system [19]. Consequently, establishing a reliable and stable NIR online analysis model for the whole process of online detection and forage maize quality control is challenging, with important practical significance. Moreover, for industrial applications, it is also essential to choose the adequate operation parameters and chemometric tools to obtain a robust model. Herein, partial least squares (PLS) was chosen to establish a model between component concentrations or properties of samples and their spectral absorbance measured at different wavelengths because of the simplicity of use, speed, and good performance [20].

In the current study, we focused on the real-time detection of forage maize quality based on a home-built grating-type NIR online system. The factor parameters (sample surface temperature, number of scans, detection optical paths, particle sizes, conveyor speeds) of the online NIR spectra acquisition were analyzed and optimized. The other two general research NIR spectrometers were chosen for the study of the reliability of the online NIR spectrometer. In addition, an NIR online multivariate analysis model for the prediction of forage maize quality (crude protein, moisture, crude ash, and energy) was established, and its reliability, applicability, and stability were analyzed and validated.

## 2. Materials and Methods

### 2.1. Sample Preparation

The experimental samples (138) of forage maize were collected from 18 provinces (Shandong, Hebei, Shanxi, Shaanxi, Sichuan, Yunnan, etc.) in China. The first batch was 108 forage maize samples for the parameter optimization and the reliability analysis of the established online model (75 samples for calibration and 33 samples for validation). The second batch was 30 samples selected from the other 110 forage maize using the Mahalanobis distance method. The raw samples passed the 8 mm standard sieve (Highway Instrument, Zhejiang, China), and the impurities such as sand and soil were removed. Then the samples were packaged in zip-lock bags and stored in a sample storage room at 20 ± 2 °C for use.

### 2.2. Experimental Analysis of Forage Maize Quality

Crude protein, moisture, crude ash, and energy were selected as the quality index of forage maize. The crude protein was detected according to GB/T 24901-2010 using an automatic Kay nitrogen analyzer (Kjeltec 2300, FOSS, Hillerød, Denmark) [21]. The moisture was measured according to GB/T 24900-2010 using an electric heating constant temperature drying oven (DHG-9123A, Shanghai Jinghong Laboratory Instrument Co., Ltd., Shanghai, China) [22]. The crude ash was tested based on GB/T 6438-2007 using a program-controlled chamber electric furnace (Shanghai Jinghong Laboratory Instrument Co., Ltd., Shanghai, China) [23]. The energy was measured using an automated oxygen bomb calorimeter (Parr 6300, PARR, Moline, IL, USA).

### 2.3. NIR Spectroscopy Systems

#### 2.3.1. Different NIR Spectrometers

The main NIR spectrometer was the home-built grating online analysis system (Appendix A) composed of a light source (two tungsten lamps), an optical splitting system (RockSolid), a detector (InGaAs), a transport component, and a sample attachment, denoted as Instrument A. The other two general research NIR spectrometers (Appendix A) were as follows: Instrument B was a SupNIR-2700 grating NIR spectrometer (Focused Photonics Inc., Hangzhou, China, instrument B) with an integrated design including a light source system, a spectral acquisition component, and a transportation component. The light source system mainly included a fiber-coupled sensor and a light source (single tungsten lamp); the spectral acquisition components were mainly composed of an interferometer (RockSolid), a detector (InGaAs), and a circuit system; the transport component was a rotating sample stage. Instrument C was a Spectrum 400 diffuse reflection Fourier transform NIR spectrometer (Perkin Elmer, Waltham, MA, USA, instrument C) with an integrated design was measured using the integrated sphere diffuse reflection method. It was equipped with an InGaAs detector and a sealed anti-vibration optical platform protecting the optical system from moisture. The transport unit was a rotating sample stage.

In addition, the spectrum acquisition software of instrument A and B was from Focused Photonics Inc., while the spectra acquired software of instrument C was from Perkin Elmer. The experimental forage maize samples were equilibrated at room temperature for 72 h before NIR spectra collection. All three NIR instruments adopted non-contact diffuse reflection.

#### 2.3.2. Spectra Acquisition

(1) Instrument A spectra collection: the sample was placed in the cuboid sample box (100 × 8 × 2 cm) on the conveyor belt, the optical detection path was adjusted to 12 cm, and the conveyor speed was set to 10 cm s^−1^. The NIR spectra were collected when the sample box was moved under the acquisition window. The sample was repeatedly loaded 3 times, and the average spectra were taken as the sample representative spectra. Spectrometer scanning parameters were as follows: the wavelength range was 1000–2500 nm, the spectral resolution was 10 nm, and the number of scans was 32.

(2) Instrument B spectra acquisition: the transport unit was a rotating sample stage. The sample was repeatedly loaded 3 times, and the average spectra were taken as the sample representative spectra. The spectrometer scanning parameters were as follows: the wavelength range was 1000–2500 nm, the spectral resolution was 10 nm, and the number of scans was 32.

(3) Instrument C spectra acquisition: the transport unit was a rotating sample stage. The sample was repeatedly loaded 3 times, and the average spectra were taken as the sample spectra. Spectrometer scanning parameters were as follows: the wavenumber range was 10,000–4000 cm^−1^, the spectral resolution was 8 cm^−1^, and the number of scans was 32.

### 2.4. Analysis of Influence Factors for NIR Online Spectrometer

#### 2.4.1. Sample Surface Temperature

The halogen tungsten lamp of the grating NIR online spectrometer used in this study generates a large amount of heat at the focus of the light, which rapidly increases the surface temperature of the sample. To study the variation of sample surface temperature and the corresponding NIR spectra during the scanning process, the spectra of forage maize samples were measured by the NIR online spectrometer, and the real-time surface temperatures of samples were detected by the MC infrared thermometer (Zhixianglingyu Technology Co., Ltd., Shenzhen, China). For each forage maize sample, eight spectra were measured at the surface temperatures of 25, 30, 35, 40, 45, 50, 55, and 60 °C. The wavelength range was 1000–2500 nm, the number of scans was 32, the detection optical path was 12 cm, and the conveyor speed was 0 cm s^−1^. The differences of average spectra of samples collected at different temperatures were compared and analyzed to study the variation of NIR spectra of forage maize with different surface temperatures.

#### 2.4.2. Different Number of Spectra Scans

Different scanning times cause variations of the sample surface temperature, which results in spectral data variation. The scanning time was decided by the different number of scans. Therefore, the number of scans was set to 1, 16, 32, and 64 to study the influence of the number of scans on the NIR spectra of the forage maize. The forage maize samples at different numbers of scans were collected and repeatedly tested 10 times.

The evaluation index of spectral reproducibility is expressed by the standard deviation (*SD*), and the calculation formula is as follows:(1)$$\;SD_{l}=\frac{\sum_{i=1}^{10}\left(A_{i}^{l}-A_{av}^{l}\right)}{10-1}$$
where $A_{i}^{l}$ is the absorbance of *l* wavelength of the *i* spectrum, and $A_{av}^{l}$ is the average absorbance of *l* wavelength.

The spectral reproducibility under different numbers of scans was calculated by the average *SD* of the representative samples. The wavelength range was 1000–2500 nm, the detection optical path was 12 cm, the environment temperature was 25 °C, and the conveyor speed was 0 cm s^−1^. The effects of different numbers of scans on the spectral reproducibility of forage maize were studied by analyzing the changes of spectral *SD*.

#### 2.4.3. Different Detection Optical Paths

The detection optical path was defined as the distance between the spectrometer scanning window and the surface of the sample. Five different detection optical paths (8, 10, 12, 14, and 16 cm) were set to evaluate the effects of different detection optical paths on NIR spectra by adding different spacers under the sample cup. The wavelength range was 1000–2500 nm, the scanning times was 32, the environment temperature was 25 °C, and the conveyor speed was 0 cm s^−1^. Using crude protein as the quality index, the quantitative analysis models of different detection optical paths were established to study the effects of different detection optical paths on the NIR spectra of forage maize.

#### 2.4.4. Different Particle Sizes

NIR spectra of raw forage maize and two other different particle size samples (1 mm and 0.425 mm) were prepared to explore the effects of different particle sizes on NIR spectra. The wavelength range was 1000–2500 nm, the detection optical path was 12 cm, the number of scans was 32, the environment temperature was 25 °C, and the conveyor speed was 0 cm s^−1^. The quantitative analysis model of crude protein with different particle sizes for forage maize was established to study the effects of different particle sizes on NIR spectra.

#### 2.4.5. Different Conveyor Speeds

The NIR spectra of forage maize samples were collected at different conveyor speeds (0, 5, 8, 10, and 20 cm s^−1^). The wavelength range was 1000–2500 nm, the detection optical path was 12 cm, the scanning times was 32, and the environment temperature was 25 °C. The quantitative analysis model of crude protein for forage maize was established at different conveyor speeds to analyze the effects of different conveyor speeds on NIR spectra.

#### 2.4.6. Orthogonal Experiment of Influence Factors of NIR Online Analysis

Based on the above-mentioned single factor experiment, the effects of the detection optical path, the conveyor speed, and the number of scans during the dynamic spectral data acquisition were further investigated using three-factor and three-level orthogonal experiments. The detection optical paths were 8, 12, and 16 cm; the conveyor speeds were 10, 20, and 30 cm s^−1^; and the numbers of scans were 8, 16, and 32 times. The orthogonal experiment of the influence factors of the NIR online analysis are shown in Appendix A.

The root mean square error (*RMS*) was used to evaluate the reproducibility of the spectrum and was calculated as follows:(2)$$RMS_{l}=\sqrt{\frac{\sum_{i=1}^{n}{\left(A_{il}-{\bar{A}}_{i}\right)}^{2}}{n}}$$
where $A_{il}$ is the absorbance of i wavelength of the l spectrum, ${\bar{A}}_{i}$ is the average absorbance of i wavelength, and n is the total wavelength point.

### 2.5. Data Processing and Analysis

An NIR quantitative analysis model was established using the PLS method in the PLS-Toolbox 8.0 based on MATLAB. Spectral preprocessing was performed by standard normal variate (SNV), detrend, and first derivative processing [10]. The NIR spectra of the forage maize were divided into the calibration set and validation set based on the concentration gradient method. Outlier samples were rejected by the studentized residual of the sample chemical value and the Mahalanobis distance [15]. The studentized residual is the ratio of the difference between the predicted value and the measured value of the sample to the standard error of the sample. The equation is calculated as follows.
(3)$$\;Y\;Stdnt\;Residual_{i}=\frac{{\tilde{y}}_{i}-y_{i}}{SEE}$$
where ${\tilde{y}}_{i}$ is the model predicted value of the *i* sample, $y_{i}$ is the true chemical value of the *i* sample, and *SEE* is the predicted standard error of all samples.

The Mahalanobis distance of a spectrum refers to the distance between the sample spectrum and the average spectrum of the entire spectrum set. After normalizing the spectral data, the Mahalanobis distance of a single sample is determined by the following equation:(4)$$h_{ii}=t_{i}^{T}{\left(T^{i}T\right)}^{-1}t_{i}$$
where $h_{ii}$ represents the impact of a single sample on the entire standard sample set. In the near-infrared spectroscopy analysis, $h_{ii}$ expresses the degree of influence of sample *i* on the regression model, $t_{i}$ is the score vector of sample *i*, and $T^{i}T$ is the score matrix of the modeling set. The grating NIR online spectrometer and two general research NIR spectrometers were chosen to compare and investigate the reliability of the NIR online analysis method for the forage maize quality. The correlation coefficient of calibration (*R_c_*), the correlation coefficient of prediction (*R_p_*), the root mean square error of calibration (*RMSEC*), and the root mean square error of prediction (*RMSEP*) were chosen to evaluate the obtained model and were expressed as follows [24].
(5)$$R_{c}=\left[\frac{\sum_{i=1}^{n_{c}}\left({\hat{y}}_{i}-{\bar{y}}_{c}\right)}{\sum_{i=1}^{n_{c}}\left(y_{i}-{\bar{y}}_{c}\right)}\right]$$
(6)$$R_{p}=\left[\frac{\sum_{i=1}^{n_{v}}\left({\hat{y}}_{i}-{\bar{y}}_{v}\right)}{\sum_{i=1}^{n_{v}}\left(y_{i}-{\bar{y}}_{v}\right)}\right]$$
(7)$$RMSEC=\sqrt{\frac{1}{n_{c}}\sum_{i=1}^{n_{c}}{\left(y_{i}-{\hat{y}}_{i}\right)}^{2}}$$
(8)$$RMSEP=\sqrt{\frac{1}{n_{v}}\sum_{i=1}^{n_{v}}{\left(y_{i}-{\hat{y}}_{i}\right)}^{2}}$$
where $y_{i}$ is the measured chemical value of the i sample, ${\hat{y}}_{i}$ is the NIR absorbance value of the i sample, $n_{c}$ is the number of samples in the calibration set, and $n_{v}$ is the number of samples in the validation set; ${\bar{y}}_{c}$ and ${\bar{y}}_{v}$ are the average chemical values of calibration set and the validation set, respectively.

The closer the *R_c_* and *R_p_* values are to 1, the closer the predicted value is to the true value. The smaller the *RMSEC* and *RMSEP* and the smaller the difference between them, the better the model performance.

Relative analysis error (*RPD*) is calculated by:
(9)*RPD* = *SD*/*RMSEP*

where *SD* is the standard deviation of the chemical values of the validation set.

The smaller the difference in model results, the more reliable the NIR online analysis.

## 3. Results and Discussion

### 3.1. Parameter Analysis of NIR Spectra Collection

#### 3.1.1. Variation of Surface Temperature and Spectral Absorption during Sample Scanning

Figure 1a depicts the variation of surface temperature of the forage maize sample against the scanning times (8 min and 1 min). The surface temperature increased continuously with the increase of scanning time. In the first minute, the rate of increase was rapid, and the fit curve of temperature against time was studied, which was Y = 0.27X + 25.86, while as time passed, the rising speed showed a decreased trend.

The NIR spectra of the forage maize samples at different temperatures are presented in Figure 1b. The spectral absorbance was the highest at 25 °C and decreased with increasing temperature. The absorption peaks at 1400–1500 nm and 1850–1950 nm corresponding to the moisture peaks varied in the intensities and positions with the changing temperature [18,25]. The temperature variation resulted in the force change between the molecules, which led to the change of vibration and rotation of the O–H, resulting in the NIR spectral shift [26]. According to a previous report [26], temperature affected the spectra in a nonlinear form. As the temperature increased, the molecular energy increased, leading to the molecular transition to higher energy levels, which contributed to the increasing absorption intensity, and the decreasing reflection intensity [27]. Consequently, the absorptions of NIR light energy of the sample molecules were weakened, and the absorption intensities were reduced. Therefore, the environment temperature and sample scanning time must be strictly controlled when collecting NIR spectral data.

#### 3.1.2. Effect of Different Number of Scans on NIR Spectral

NIR spectra and *SD* values of the forage maize samples with different numbers of scans are presented in Figure 1c, as well as the NIR spectra (*SD* values) with SNV and detrend preprocessing (Figure 1d). For the original spectra, the spectral noise decreased as the number of scans increased. Comparing the NIR spectra with different numbers of scans, the *SD* value was the highest under the number of scans of 1 and decreased until the number of scans increased to 32 and then gradually stabilized. The NIR spectra were further preprocessed by SNV and detrended; however, the spectral noise cancellation was not obvious for the spectra with different numbers of scans.

Based on the above mentioned, increasing the number of scans increased the signal-to-noise ratio; however, the increase of number of scans led to the extended scan time and the pronounced change in the sample surface temperature, which thereby caused the NIR spectra shift [18]. Therefore, the reasonable selection of the number of spectra scans can effectively reduce the influence of temperature change on the spectra, as well as improve the detection efficiency in the online process. According to the results, the number of scans of 32 was considered the reasonable parameter at which the spectral data were relatively stable, and the detection efficiency was high.

#### 3.1.3. Effect of Different Detection Optical Paths on NIR Spectra

Figure 1e presents the NIR spectra of forage maize with different detection optical paths in which the spectra with detection paths of 8 cm and 10 cm were overlapped, while the other spectra showed obvious differences with different detection paths. The spectral absorbance increased with the increase of the detection optical path. The NIR spectra with different detection optical paths were further analyzed by principal component analysis (PCA) and are shown in Figure 1f. Most samples at the detection optical path of 8 cm and 10 cm were overlapped, and they partly overlapped with the sample of the 12 cm detection optical path, while the samples with the detection optical path of 12 cm and 16 cm presented the obvious difference, which indicated the difference among the forage maize spectra that presented an increasing trend with the increase of the detection optical path.

Appendix A lists the results of the quantitative analysis model for crude protein of forage maize at different detection optical paths. The selected optimal preprocessing method was the SNV, detrend, and first derivative processing [20]. The results revealed that the NIR quantitative model of crude protein at the optical path of 12 cm was ideal (RPD = 2.32), in which the coefficient of calibration and prediction were both larger than those in other detection optical paths. As the detection optical path decreased or increased, the prediction effect of the model was reduced.

#### 3.1.4. Effect of Different Particle Sizes on NIR Spectra

The NIR spectra of forage maize with different particle sizes are depicted in Figure 1g, which revealed there are distinct differences between the raw forage maize and the crushed samples (1 mm and 0.425 mm). The spectral absorbance increased with the increase of the particle size. This is because the change of particle size affects the light penetration and reflection characteristics, which contributes to the light propagation in the sample variation to some extent and influences the scattering and absorption of light by the sample. In addition, the scattering coefficient of the sample decreased as the particle size increased; thus, the larger the particle size, the higher the absorbance of the overall spectra [19]. The PCA results of NIR spectra with different particle sizes (Figure 1h) illustrated that the principal component score of different particle size presented differences. The smaller the difference in the particle size, the smaller the difference in the performance of the principal component score [27].

Appendix A shows the results of the quantitative analysis model for crude protein of forage maize in different particle sizes. The *R_c_* of the crude protein analysis model established by NIR online spectra were all greater than 0.94, and the RPD were higher than 2.32. The prediction of the quantitative model for 0.425 mm crushed maize was better than the predictions for the raw maize and 1 mm crushed maize.

#### 3.1.5. Effect of Different Conveyor Speeds on NIR Spectra

The results of the quantitative analysis model for crude protein of forage maize at different conveyor speeds are displayed in Appendix A. The crude protein model at 5 cm s^−1^ was better than those at 10 and 20 cm s^−1^, of which *R_c_* and *R_p_* were 0.94 and 0.86, respectively, *RMSEC* and *RMSEP* were 0.22 and 0.31, respectively, and RPD was 1.87. With the increase of conveyor speed, the performance of the prediction model for crude protein was decreased but not distinct.

Treating the spectra at 0 cm s^−1^ as the standard spectra, the spectral residuals between the spectra at 5, 10, and 20 cm s^−1^ and the standard spectra were considered as noise. Figure 1i–l depicts the residual distribution and residual probability density distribution without and with SNV + detrend + first derivative preprocessing. According to the residual distribution, noises at different wavelengths were different. However, SNV + detrend + first derivative preprocessing significantly reduced the noise [11,20]. Residual probability density distribution after spectral preprocessing also conformed to the Gaussian distribution; therefore, spectral preprocessing could effectively reduce the influence of speed noise.

Appendix A shows the signal-to-noise ratios at different conveyor speeds. The signal-to-noise ratios of water, crude protein, and crude fat peaks gradually decreased with increasing speed. This was consistent with the results in Appendix A showing that the quantitative model for crude protein of forage maize declined as the speed increased.

The change rate of the spectral area is also the evaluation index of spectral stability [14]. Appendix A presents the change rate of the spectral area under different conveyor speeds, in which the change rate at the condition of 5 cm s^−1^ was the smallest (0.003). As the speed increased, the change rate of the spectral area increased continuously. If there is no noise in the spectra, the spectra collected under the same conditions should be coincident [18]. That is, the smaller the change rate of the spectral area, the more stable the spectral detection system. Therefore, as the conveyor speed increased, the spectral noise increased, and the result of the quantitative analysis model for crude protein of forage maize continued to decline.

#### 3.1.6. Parameter Optimization of Influence Factors of NIR Online Analysis

The parameter combinations of orthogonal experiments for the different influence factors of NIR online analysis are listed in Appendix A, which also reveals the different *RMS* values. The results of multivariate analysis of variance are presented in Appendix A. The number of spectra scans and the detection optical path were both significant at the significance level of 0.05 (F > F0.95(2,2) = 19), while the conveyor speed was not significant (F = 0.633 < F0.95(2,2) = 19). Furthermore, the significance of the number of spectra scans was greater than that of the detected optical path (107.985 > 46.848).

Based on the above mentioned, when the detection optical path was 12 cm, the conveyor speed was 10 cm s^−1^, and the number of spectra scans was 32, and the *RMS* value was the smallest, that is, the spectral reproducibility was the best, which can be used as the optimal parameter reference for NIR online dynamic spectral data acquisition [14].

### 3.2. Establishment and Comparison of Forage Maize Model Using Three Different NIR Spectrometers

#### 3.2.1. Component Analysis of Forage Maize

The statistical results of crude protein and moisture content of experimental forage maize are presented in Table 1, the averages of which were 8.01% and 12.18%. The crude protein content ranged from 5.44% to 10.76%, and the variance was 0.79; the moisture content ranged from 7.16% to 15.87%, and the variance was 1.84. According to the distribution of crude protein and moisture content (data not shown), protein and moisture both followed a normal distribution [28].

#### 3.2.2. Spectral Characteristics of Different NIR Spectrometers

The spectra of forage maize collected from three different NIR spectrometers are depicted in Figure 2, which reveals there were some differences in the spectral absorbance among the three instruments. The frequency multiplication and frequency combination of the atom vibration in the molecule mostly occurred in the NIR region. As seen in Figure 2, the NIR original spectra of maize had three distinct characteristic peaks, namely, 1196 nm, 1450 nm, and 1930 nm. The peaks at 1196 and 1450 nm corresponded to the second-order vibration of the O–H bond, and the characteristic peak at 1930 nm corresponded to the first-order vibration of the O–H bond, all of which were mostly related to the moisture content of maize [7].

Figure 2d displays the NIR average spectra (second derivative) of forage maize collected by three different spectrometers. The peak around 1700–1735 nm was mainly the C–H first-order vibration, which is the lipid absorption caused by the structure of CH_2_ and CH_3_ in maize oil; the peak at 2248 nm was the combined vibration of C–H and N–H, mainly due to the absorption of crude protein; the peaks at about 2330 nm and 2349 nm were C–H stretching vibration and deformation vibration, which are related to the fiber composition of maize [18,29].

#### 3.2.3. Comparative Analysis of Models of Different NIR Spectrometers

The crude protein and water models of raw forage maize for different NIR spectrometers were established using PLS combined with spectral preprocessing (SNV, detrend, and first derivative). The model parameters and the scatter plots are presented in Figure 3. Comparing the optimal quantitative models of crude protein established by three NIR spectrometers, the *R_p_* and *RMSEP* of instrument A, B, and C were 0.78 and 0.40, 0.86 and 0.31, and 0.86 and 0.36, respectively. The crude protein results illustrated that the instrument B and C performed slightly better than instrument A, but the difference was not significant (data not shown). As for the moisture, the models of the three instruments were similar.

The SupNIR-2700 grating NIR spectrometer (instrument B) and Spectrum 400 diffuse reflection Fourier transform NIR spectrometer (instrument C) are market-oriented products, and the accuracy and stability have been widely recognized in different industries. Comparing the three NIR spectrometers, the quantitative analysis model of forage maize established by the NIR online spectrometer (instrument A) was basically the same as the other two general research NIR spectrometers, indicating that the prediction of the NIR online analysis spectrometer was reliable and meets the needs of practical applications.

As seen in Figure 3, the crude protein model was better than the moisture model, which is because the moisture of maize particle did not appear evenly distributed. From the outer to the inner of the grain, the moisture content was reduced in turn, and the moisture content was also easily affected by the external environment or the temperature compared with the crude protein [30]. Therefore, the crude protein spectral information was superior to the moisture spectral information when acquiring the NIR spectra.

### 3.3. Validation Analysis of NIR Online System for Forage Maize Quality Detection

#### 3.3.1. Component Statistics of Forage Maize

Table 1 lists the contents of crude protein, moisture, crude ash, and energy of raw and crushed (0.425 mm) forage maize. Some outliers have been removed due to the abnormal values. In raw samples, 5, 6, 4, and 5 samples were rejected for crude protein, moisture, crude ash, and energy, respectively, while in the crushed sample, 0, 7, 4, and 0 samples were respectively removed. As seen in Table 1, the component contents between the raw and crushed forage maize were slightly different, which may be caused by the mechanical milling process.

#### 3.3.2. Establishment of NIR Online Model for Forage Maize Quality

The NIR online multivariate analysis model of raw and crushed forage maize was established by PLS combined with spectral processing (SNV, detrend, and first derivative). The model parameters and scatter plots are depicted in Figure 4. The *R_c_* and *R_p_* of models for the crude protein, moisture, crude ash, and energy of the raw forage maize were all lower than those of the crushed forage maize, while the *RMSEC* and *RMSEP* of model for the raw maize were greater than those of the crushed maize, which illustrated that the prediction of the model for the crushed forage maize performed better than that for the raw forage maize, which was also in accordance with the results from the quantitative analysis model of crude protein of forage maize with different particle sizes [31]. Comparing the model results of the crude protein, moisture, crude ash, and energy, the crude protein model was the best, the moisture and energy model were slightly worse, and the crude ash model was not satisfied, mainly because of the low crude ash content and the uneven distribution in the forage maize.

### 3.4. Verification for Practical Application of NIR Online Model for Forage Maize Quality

#### 3.4.1. Repeatability Analysis of NIR Online Model for Forage Maize

Based on the dynamic online analysis system, the representative 10 forage maize samples were continuously and repeatedly recorded 20 times to obtain the mean values of crude protein, moisture, crude ash, and energy. The plots of prediction results over time for the raw and crushed forage maize were established and are presented in Appendix A. According to the real-time online spectral data within 4 min, the dynamic prediction results of crude protein, moisture, crude ash, and energy of forage maize fluctuated around the true values of 7.98, 12.22, 1.10, and 16.72, respectively, indicating that the prediction ability of the model had good repeatability [28,31].

In addition, the fluctuation of dynamic prediction results for the crude protein, moisture, crude ash, and energy of crushed forage maize were significantly less obvious than those of raw forage maize, which illustrated the prediction reproducibility of the crushed maize model was much better than that of the raw maize model. This might be because the crushed sample was more homogenous compared with the raw sample, which resulted in a more stable spectra being generated by the crushed sample.

#### 3.4.2. Accuracy Analysis of NIR Online Model for Forage Maize Quality

Table 2 shows the comparison between the true values of the forage maizes (raw and crushed sample) and their NIR predicted values. According to the statistical result of the representative raw sample, the relative errors of most crude proteins were around 0.3%; the relative errors of all moisture were below 3%; the relative errors of crude ash were all greater than 15%, mainly due to the low crude ash content and non-uniform distribution in forage maize; furthermore, the relative errors of energy were all lower than 2.5%.

From the statistical results of the representative crushed samples, the relative error of crude protein and moisture were all less than 2.5%; the relative error of energy was lower than 1.1%, all of which were satisfied with the relative error requirement. However, the relative error of crude ash was not as good as the other components. Based on the statistical analysis of repeatability and accuracy of model prediction, NIR provides an effective data reference for the quality control of forage maize in practical production process [28].

## 4. Conclusions

In the current study, a home-built grating online analysis system was compared with the other two general research NIR spectrometers. The number of spectra scans, particle size, detection optical path, and conveyor speed were analyzed to investigate the influence factors and parameter optimization. The results indicated that a detection optical path of 12 cm, a conveyor speed of 10 cm s^−1^, and number of scans of 32 were the optimal parameters. Moreover, the decrease of particle size effectively improved the model performance. Choosing the crude protein and moisture of forage maize as quality indicators, the home-built grating online analysis system (instrument A) and two general research NIR spectrometers (grating instrument B, Fourier transform instrument C) were to establish the quantitative analysis model of forage maize. According to the characteristics of spectral data and the results of the quantitative analysis model, the reliability of the home-built NIR online analysis system was confirmed. Based on the home-built grating NIR online spectrometer, NIR online multivariate analysis models for the raw and crushed forage maize were established. The correlation coefficients of validation for crude protein, moisture, crude ash, and energy of the crushed forage maize were slightly greater than those of the raw maize, both of which were proven to be suitable for online practical application. Online dynamic verification illustrated that the model prediction of the home-built NIR online analysis system had good accuracy and repeatability.

## Figures and Tables

**Figure 1 foods-11-03490-f001:**
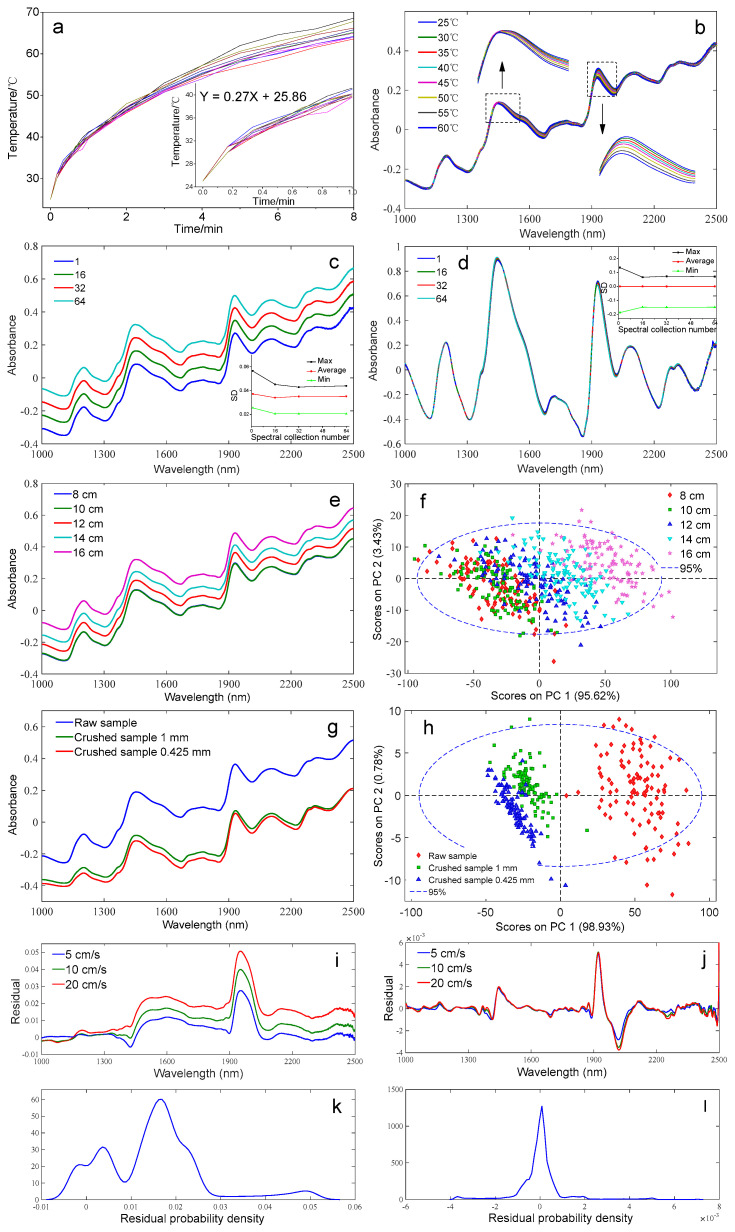
NIR results at different sample surface temperatures, number of scans, detection optical paths, particle sizes, and conveyor speeds ((**a**) the variation of surface temperature of the forage maize against the scanning time; (**b**) NIR spectra of the forage maize at different temperatures; (**c**,**d**) NIR spectra (*SD* values) of the forage maize with different number of scans without and with SNV and detrend preprocessing; (**e**,**f**) NIR spectra and PCA result of the forage maize with different detection optical path; (**g**,**h**) NIR spectra and PCA result of the forage maize with different particle sizes; (**i**−**l**) the residual distribution and residual probability density distribution without and with SNV + detrend + first derivative preprocessing).

**Figure 2 foods-11-03490-f002:**
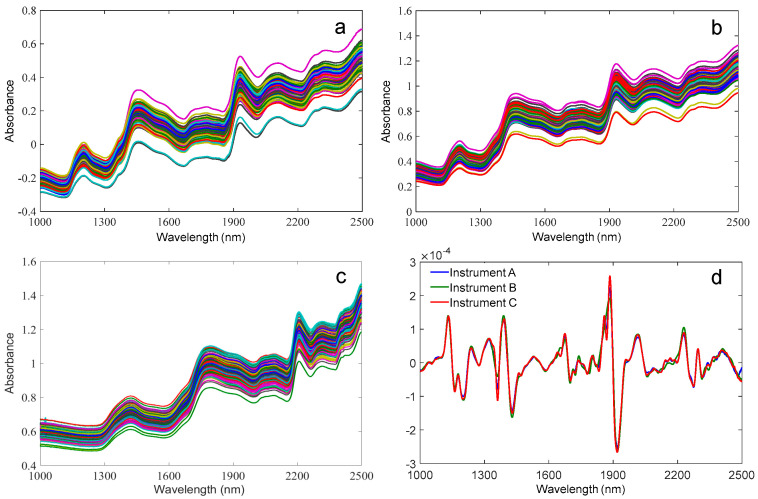
Spectra of forage maize collected from three different NIR spectrometers and their representative spectra of the second derivative ((**a**) instrument A; (**b**) instrument B; (**c**) instrument C; (**d**) NIR average spectra (second derivative) of forage maize).

**Figure 3 foods-11-03490-f003:**
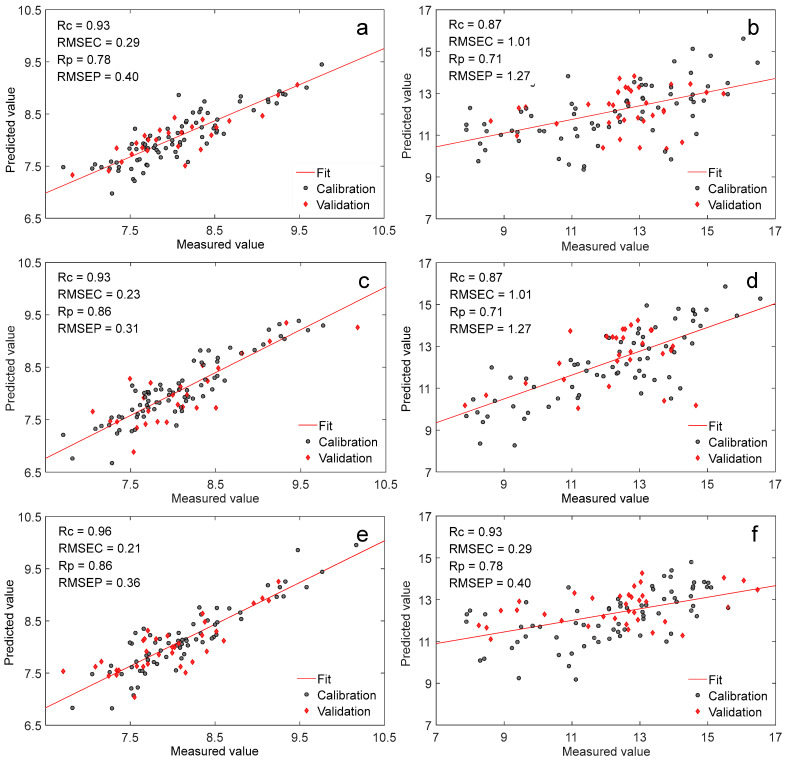
Scatter plots of crude protein and moisture models of forage maize for different NIR spectrometers ((**a**,**c**,**e**) crude protein model of instrument A, B and C; (**b**,**d**,**f**) moisture model of instrument A, B, and C).

**Figure 4 foods-11-03490-f004:**
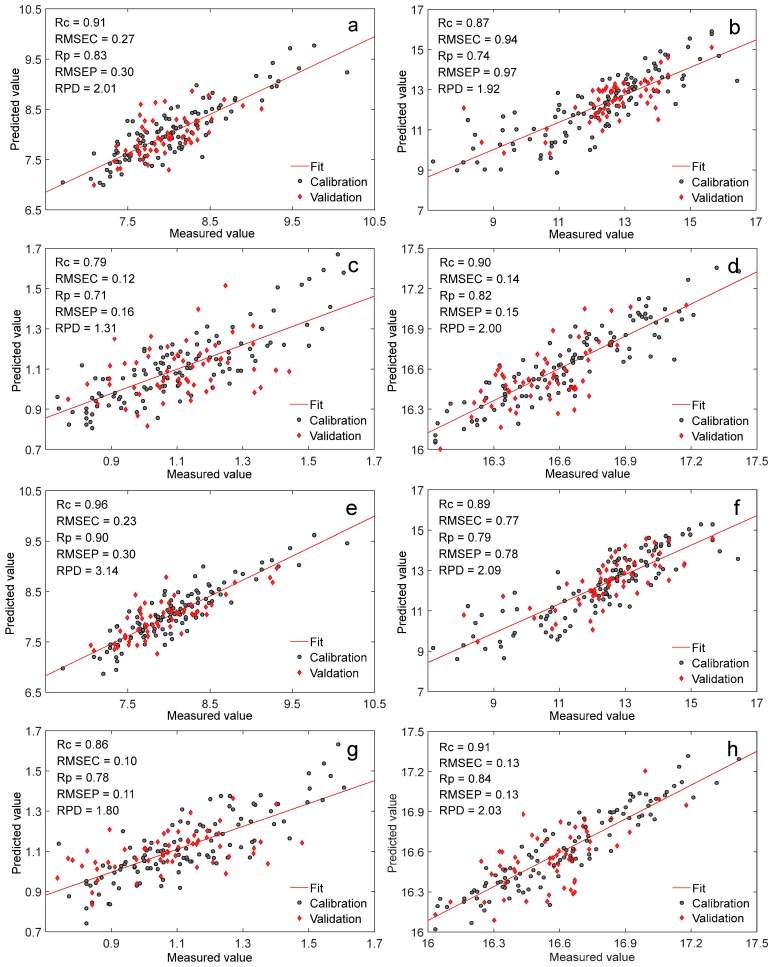
Scatter plots of NIR online analysis model for raw and crushed forage maize by the home-built spectrometer ((**a**,**c**,**e**,**g**) crude protein, moisture, crude ash, and energy models of raw forage maize; (**b**,**d**,**f**,**h**) crude protein, moisture, crude ash, and energy models of crushed forage maize).

**Table 1 foods-11-03490-t001:** Component statistics of raw and crushed forage maize samples.

Sample		Calibration				Validation			
		Maximum Value (%)	Minimum Value (%)	Average (%)	Variance	Maximum Value (%)	Minimum Value (%)	Average (%)	Variance
Raw sample	Crude Protein	10.76	5.44	8.01	0.79	8.52	5.44	7.45	0.60
	moisture	15.87	7.16	12.18	1.84	16.42	10.88	13.24	1.86
	Crude ash	1.61	0.74	1.11	0.19	1.40	0.77	1.05	0.21
	Energy	17.42	15.94	16.61	0.31	17.08	15.98	16.44	0.30
Crushed sample	Crude Protein	10.76	5.44	7.95	0.78	9.25	5.44	7.67	0.94
	moisture	16.42	7.16	12.25	1.83	15.65	9.27	12.90	1.63
	Crude ash	1.61	0.77	1.12	0.19	1.48	0.74	1.03	0.19
	Energy	17.42	15.94	16.59	0.31	17.01	15.98	16.50	0.26

**Table 2 foods-11-03490-t002:** Comparison of experimental values and NIR predictions of raw and crushed forage maize samples.

		Crude Protein (%)			Moisture (%)			Crude Ash (%)			Energy (MJ kg^−1^)		
		True Value	Predicted Value	Relative Error	True Value	Predicted Value	Relative Error	True Value	Predicted Value	Relative Error	True Value	Predicted Value	Relative Error
Raw Sample	1	8.43	8.33	1.18	9.03	9.30	2.99	1.04	0.86	17.3	17.02	16.86	0.94
	2	8.09	7.88	2.59	12.37	12.65	2.26	1.18	0.96	18.6	16.74	16.38	2.15
	3	7.88	7.64	3.05	11.75	11.50	2.13	1.18	0.98	16.9	16.66	16.53	0.72
	4	7.87	7.66	2.67	12.62	12.38	1.90	1.08	0.84	22.22	16.77	16.40	2.21
	5	7.59	7.84	3.16	12.94	12.80	1.08	1.02	0.86	15.68	16.59	16.57	0.12
	6	7.77	7.55	2.83	13.26	12.94	2.41	1.01	0.79	21.78	16.53	16.36	1.02
	7	7.98	7.72	3.26	13.21	12.92	2.20	1.08	0.88	19.44	16.64	16.36	1.68
	8	7.89	8.12	3.04	12.81	12.54	2.03	1.10	0.93	15.45	16.74	16.33	2.45
	9	7.74	7.54	2.58	13.12	13.17	0.38	1.14	0.85	25.43	16.63	16.40	1.38
	10	8.58	8.34	2.80	11.04	10.90	1.27	1.19	0.97	18.5	16.90	16.85	0.35
Crushed sample	1	8.43	8.40	0.35	9.03	9.00	0.33	1.04	1.00	3.85	17.02	16.90	0.71
	2	8.09	8.27	2.22	12.37	12.63	2.10	1.18	1.23	4.24	16.74	16.63	0.65
	3	7.88	7.96	1.01	11.75	11.50	2.13	1.18	1.13	4.24	16.66	16.56	0.60
	4	7.87	8.04	2.03	12.62	12.79	1.43	1.08	1.24	14.8	16.77	16.59	1.07
	5	7.59	7.76	2.24	12.94	12.91	0.23	1.02	1.15	13.7	16.59	16.48	0.66
	6	7.77	7.96	2.45	13.26	13.05	1.58	1.01	1.22	20.7	16.53	16.42	0.72
	7	7.98	8.15	2.13	13.21	13.42	1.59	1.08	1.13	4.6	16.64	16.48	0.90
	8	7.89	8.10	2.66	12.81	12.54	2.03	1.10	1.20	0.09	16.74	16.60	0.83
	9	7.74	7.90	2.07	13.12	13.17	0.38	1.14	1.09	4.38	16.63	16.48	0.90
	10	8.58	8.78	2.33	11.04	11.09	0.45	1.19	1.21	1.68	16.90	16.81	0.53

## Data Availability

The data presented in this study are available on request from the corresponding author.

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
