# Peer review of "A Study of the Reliability and Accuracy of the Real-Time Detection of Forage Maize Quality Using a Home-Built Near-Infrared Spectrometer"

_foods, 2022, doi:10.3390/foods11213490_

Round 1

Reviewer 1 Report

In the present manuscript, the authors aim at validating the use of a homemade NIR spectrometer for monitoring the forage maize quality.

The topic addressed by the authors is interesting, but the manuscript is not well-organized and presents main inaccuracies. In addition, some technical terms are not correctly used. The numerous imprecisions in the manuscript make it difficult to assess the quality of the analyzes reported by the authors. For these reasons, the paper should be deeply revised to meet the journal's quality standards.

In the following, a non-exhaustive list of the various points that should be addressed by the authors is given.

a)      Please check the title of the paper. Probably the authors would present a home-built spectrometer not “a home-build spectroscopy”. The contemporary use of “online” and “dynamic” should be substantiated.

b)      Also, the highlights should be revised. What do the authors exactly mean by “influence factors”? The terms “Model prediction of home-built NIR online system” are not very meaningful in the highlights.

c)      There are a certain number of terms that are not clear, i.e. The parameters of factors, scan times, spectral collected software, spacious difference etc.

d)      Check the unit of measurement of the optical path, which sometimes is given as cm s-1 (see for instance line 33).

e)      The reference given in line 87 is very old. Please update the literature concerning the sample motion's influence on NIR measurements.

f)       Since the paper's main aim is to validate a home-built spectrometer, the description of all the components in the 2.3. 1 paragraph should be more accurate (product number, manufacturers and so on). A schematic layout of the system would be useful. As far concerns the two commercial systems references to the technical notes of the manufacturers should be given.

g)      What was the criterion with which the authors chose the two different sizes of the samples?

h)      As far as concerns the data analysis procedures, please add some references to help the reader in evaluating this aspect.

i)        In Equations 5 and 6, the authors introduced “the NIR detection value” but they do not define what is exactly this value in the different examined cases.

j)        Please add the error in the parameters obtained from the different performed fitting procedures.

k)      In paragraph 3.1.1 please clarify why the regions 1400-1500 nm and 1850-1950 nm are assumed to be related to moisture. Proper references could be useful for substantiating these statements and the ones related to the effects induced by the change in temperature (lines 325-335).

l)        Please revise lines 393-395, the effect of the size should be better discussed.

m)   What exactly do the authors mean by “crude protein model”?

n)      In Figure 2 the horizontal axis should be similar for the a, b, and c panels.

Reviewer 2 Report

In the present work the authors study the reliability and accuracy of the real-time detection for forage maize quality using the home-built dynamic online near-infrared
spectroscopy. For it, the obtained results were compared with those obtained using other two general research NIR spectrometers.
It could be considered useful and interesting for the industry. The paper is well organized and the obtained results are interesting; nevertheless, there are some aspects that have to be considered. For the above reasons I suggest major revision and reconsideration.

Line 108: The parameters that will be used in this work to define the quality of the forage maize should be detailed.

Line 118: what criteria was used exactly? What characteristics did the 30 samples that were used have that the rest did not? This aspect needs to be explained in more detail. How many samples constitute the calibration group and how many the validation group? Specify in the text

Line 125-135: A more detailed explanation of the different experimental determinations is needed. references?

Line 143: Be careful how you refer to the different instruments. You have previously quoted them as a, b and c but in this line you call b 1. However, in the next section 1 is a. The nomenclature must be unified to avoid confusion.

Line 165-168: The wording should be changed according to the style of the rest of the work. The imperative is not used, it describes what has been done.

Line 170: three or 3? unify criteria throughout the paper

Line 276-278: Why are these pretreatments used? Have others been tried?

Line 280: it should be explained what the concentration gradient method is. ¿reference?

Line 281: Explain the method and the specific criteria for considering outliers. On numerous occasions throughout the work, the authors simply name a method or procedure without giving detailed explanations or references to it. Please revise.

Line 371: 12 or 14?

Line 416: cm s-1 or cm/s? unify throughout the manuscript, including tables.

Line 459: Has it been evaluated whether the interaction between the different parameters has influence on the result?

Line 516: raw or crushed? You mention the two types in the work, specify which has been used in this case

Line 555: 1mm or 0.425 mm? Specify

Line 556: What are abnormal values?

Line 557: The samples that have been eliminated correspond to which group, calibration or validation?

Table 1 and 2: adjust the font or table size.

Table S-4. The table must have a table footer specifying that it is A, B, C, etc. Check the rest of the tables, none has a foot

Fig. S1-a and b have very low resolution. Especially in a, it is impossible to see the situation of the different components that are detailed in the text.

Round 2

Reviewer 1 Report

In my opinion, the authors significantly improved their manuscript, but some further attention should be given to the following points:

1)     “Scan times” should be more properly indicated as “number of scans”

2)     As I said before the paper's main aim is to validate the home-built spectrometer. Even though the authors add some details, the description of the instruments is still not complete (e.g., It is not sufficient to write InGaAs for the detector, please give its product number, manufacturers and so on. Please do the same for all the components). The presented schematic layout of the system is not so clear, the optical path of the infrared light inside the instrument should be illustrated.

3)     The criterion with which the authors chose the two different sizes of the samples should also be given in the manuscript.

4)     As said before, please add the technical notes of the manufacturers for the two commercial systems.

5)     The discussion about regions 1400-1500 nm and 1850-1950 nm that are assumed to be moisture-related is still unclear. In addition to the proper references that have been added, some additional comments would be useful for readers.

6)     Please check the following sentences (at lines 348-353) because they are really unclear. “The temperature increases, leading to the increase of molecular energy. The probability of molecular transition to higher energy levels increases, which contributes to the absorption intensity increases, and the reflection intensity decreases. Consequently, the absorption of NIR light energy by the sample molecules was weakened, and the absorption intensity was reduced.”

7)     In the discussion of the effects of the size please clearly indicate that the amount of examined samples was the same in all the examined experimental situations. Otherwise, different amounts of the sample obviously give spectra with different intensities.

8)     Sorry, but the explanation given for the “crude protein model” is not clear to me. Perhaps, for “crude protein” would like the authors to intend the total content of proteins in the samples? Please clarify this point.

9)     Some misprints and misuse of English terms are still present. A further revision of the English language would be beneficial for the manuscript.

Reviewer 2 Report

-

Author Response

We sincerely appreciate the reviewer's affirmation to our work.